# Evaluation of a game-based training course to build capacity for insecticide resistance management in vector control programmes

Claire Dormann[1][¤], Kirsten Duda[1], Busiku Hamainza[2], Delenesaw Yewhalaw[3], Charlotte Hemingway[4], Marlize Coleman[5], Michael Coleman[1], Edward Thomsen[1]*

1 Department of Vector Biology, Liverpool School of Tropical Medicine, Liverpool, United Kingdom, 2 National Malaria Elimination Centre, Ministry of Health, Lusaka, Zambia, 3 Department of Biology, College of Natural Sciences, Jimma University, Jimma, Ethiopia, 4 Department of International Public Health, Liverpool School of Tropical Medicine, Liverpool, United Kingdom, 5 Innovative Vector Control Consortium, Liverpool, United Kingdom

¤ Current address: Department of Computer Science, University College London Engineering Science, London, United Kingdom
* edward.thomsen@lstmed.ac.uk

**Data Availability Statement:** All relevant data from the quantitative analysis are within the manuscript and its Supporting Information files. For the

## Abstract

Across Africa, malaria control programmes are increasingly challenged with the emergence of insecticide resistance among malaria vector populations. Confronted with this challenge, vector control staff must understand insecticide resistance management, think comprehensively and react positively when confronted with new problems. However, information on the subject is often only available through written guidelines that are difficult to put into practice. Based on the successes and strengths of educational games for health, we developed and evaluated a novel game-based course to fill the gap in training resources for insecticide resistance management. The training was evaluated by analysing results of pre- and post-course knowledge tests and self-efficacy surveys, as well as post-course interviews. At the start of the training, fundamental concepts of insecticide resistance were reviewed through Resistance101, a mobile app game. Subsequently, insecticide resistance management strategies were explored using the simulation game ResistanceSim, which was introduced by mini-lectures and complemented by class discussions and group work. The game-based training was conducted and evaluated in two African countries (Ethiopia and Zambia) using a mixed-methods approach. Quantitative outcome measures included knowledge acquisition and change in self-efficacy. We completed a qualitative inductive thematic analysis of participant interviews to explore the views and experiences of participants with the games and training, and the impact of the training on professional practices and attitudes. The game-based training increased knowledge in the short-term and improved self-efficacy scores. The training increased participants' knowledge base, stimulated knowledge sharing and changed work practices. The game-based training offers scalable training opportunities that could nurture and capacitate the next generation of professionals in vector control.

qualitative analysis, the paper used a "minimal data set" consisting of illustrating quotes and the final code for themes and sub-themes. However, the underlying raw data (interview transcripts) contain potentially identifying contextual and sensitive information, thus is not available. Any release of data to other parties will likely require a new ethics approval. Requests by qualified researchers to access our raw data may, therefore be addressed to LSTM Research Ethics Committee: lstmrec@lstmed.ac.uk.

**Funding:** Support for the development of Resistance101 and the development/evaluation of the game-based training was provided by the European Commission H2020 (https://ec.europa. eu/programmes/horizon2020/en) project DMC-MALVEC (GA-688207). This grant was awarded to John Vontas with MiC and MaC as co-Investigators. Funding for development of ResistanceSim and support for our external scientific advisory committee was provided by IVCC (www.ivcc.com) under the grant OPP1148615 from the Bill & Melinda Gates Foundation (www.gatesfoundation.org), of which a sub-award was awarded to MaC. The funders had no role in study design, data collection and analysis, decision to publish, or preparation of the manuscript.

**Competing interests:** I have read the journal's policy and the authors of this manuscript have the following competing interests: MaC made substantial contributions to study conceptualisation. At this time, MaC was a researcher working for the Liverpool School of Tropical Medicine. Before study implementation, MaC became employed by IVCC, who supported the development of ResistanceSim. While at IVCC, MaC did not contribute to study design or implementation, and her involvement was limited to reviewing, editing, and approving the manuscript. This does not alter our adherence to PLOS ONE policies on sharing data and materials.

## Introduction

Malaria is the second most common cause of infectious disease deaths in the world and is one of the greatest global health challenges today. It affects millions of some of the most vulnerable people living in tropical and subtropical regions, especially sub-Saharan Africa which has 90% of the malaria cases [1].

Effective control of *Anopheles* mosquitoes, the vectors of malaria, is essential to the global malaria prevention strategy and relies primarily on insecticide-based interventions. The prolonged and repetitive use of insecticides has selected for resistance globally, making insecticide resistance one of the greatest threats to sustainable vector control. The World Health Organisation issued a guidance document, "The Global Plan for Insecticide Resistance Management," outlining the strategies that should be used to mitigate the effects of resistance and prevent its spread [2]. Country teams are responsible for developing and implementing their own strategies based on the guidance, but this requires that malaria vector control staff understand the biology of resistance and how to adapt guidance to their local context. Currently, there is no standardised training or capacity strengthening package available to support control programmes in developing insecticide resistance management (IRM) plans.

Recent studies have identified the benefits of using educational games for societal, health and environmental issues [3–5]. Educational games are well-known to enhance players' motivation, creating conditions that facilitate learning and enhance competencies to empower adult learners [6–7]. They also facilitate the understanding of complex systems, overcoming fatigue and disengagement with traditional "chalk and talk" training, thus increasing the retention of content [8]. More specifically, simulation games are "experimental, rule-based, interactive environments, where players learn by taking actions and by experiencing their effects" [9, p. 125]. They allow players to develop their own strategies, try out different alternatives, and explore the implications of their decisions, without real-world consequences.

ResistanceSim is a simulation game that represents the key components and challenges of an IRM programme [10]. Resistance101 is a mobile app that teaches people about insecticide resistance in malaria vectors. We developed a game-based training course using the two games to build IRM capacity among malaria vector control staff. We completed a comprehensive evaluation of the game-based training to fully understand if and how it facilitated knowledge acquisition and whether it impacted participants' attitudes and work practices.

## Materials and methods

We conducted and evaluated the game-based training course in two African countries: Ethiopia and Zambia. The countries were chosen to represent a range of malaria transmission and insecticide resistance scenarios, as well as a diverse set of stakeholders involved in malaria control. The course used the two educational games, ResistanceSim and Resistance101, as well as mini-lectures, class discussions and group work. The evaluation of the game-based training was based on a mixed-methods approach.

### Description of the game-based training course

Resistance101 [11] is an app-based game that covers the fundamental principles of insecticide resistance and is freely available on Google Play and App Stores. The game slowly introduces new concepts to facilitate the player's understanding of the content. There are eight key learning areas including insecticide-based vector control, resistance and its spread, insecticide classes and modes of action, resistance mechanisms, multiple resistance and insecticide resistance management strategies. Each key learning area consists of an educational animation explaining scientific concepts, followed by a series of game levels that allow the players to apply those

concepts. At the end of each area, the main concepts are summarised through dynamic flash cards. For each game level, the player's mission is to choose the correct insecticide and release it at the optimal time from an intervention zone (Fig 1) to eliminate the vector population. As the game progresses and resistance becomes more complex to manage, players must choose specific combinations or sequences of insecticides to succeed.

ResistanceSim was developed through a user-centred, agile software development approach [10]. On accessing ResistanceSim, players are directed to a RoadMap, a guided walkthrough of the game functioning as a tutorial and introduction to IRM. The RoadMap is divided into missions, each one with specific learning objectives corresponding to fundamental steps in establishing an IRM plan. Once participants have completed the RoadMap, they move to the Open Simulation where players then oversee a vector control programme situated in four different districts. To succeed, players are required to manage resources and develop suitable strategies for IRM following best practices.

The training took place over 2.5 days and was facilitated by one of the co-authors (ET). The course started with an introduction to IRM through a mini-lecture and a group discussion, followed by an overview of the course. Participants were then introduced to Resistance101 and played the game for three hours, followed by a short discussion of the gameplay. Participants played the games on tablets (Lenovo tab 3) supplied by the trainers. The following day, the participants were introduced to ResistanceSim and completed the RoadMap. For each step of the IRM process, concepts were first introduced through mini-lectures, and then participants completed the corresponding missions in the RoadMap. On the final day, to support the transition from the RoadMap to the Open Simulation and strengthen their understanding of IRM, participants were asked to work in groups to manage the IRM process in one district. Each group was assigned a different IRM strategy to implement. After the Open Simulation game play session, participants presented their results and discussed the different strategies, problems and issues. At the end of the training, participants were provided with copies of the games and encouraged to continue playing on their own.

## Evaluation methods

The evaluation of the game-based training was based on a mixed-methods approach. A within group before-after design was employed with the evaluation instruments completed at the beginning and end of the course. We conducted follow up interviews with the participants one month after the course. Quantitative measures were used to assess knowledge acquisition and self-efficacy. Qualitative methods were used to explore the views and experiences of participants on the games and training, and the impact of the course on their behaviour in the work environment.

**Participants and recruitment.** Twenty-three participants from Zambia and twenty-five participants from Ethiopia were recruited by purposive sampling by co-authors familiar with the vector control stakeholders in each country (DY and BH). Participants were employees of government, universities, or research institutes, whose work related to vector control. Participants were introduced to the study in the first instance via email. One day prior to the start of the course, we gave participants an information sheet and a copy of the consent form. We verbally described why the study was being conducted, the methods used, and benefits and risks to the participants. Any questions or concerns were discussed. Participants were then allowed to consider their participation before providing written consent the following morning.

**Evaluation instruments.** To assess knowledge acquisition, we developed a multiple-choice questionnaire. It was designed to assess different levels of cognitive learning based on Bloom's taxonomy, from recalling factual knowledge to synthesising different pieces of information to solve a problem [12]. The knowledge test included 22 questions about insecticide resistance

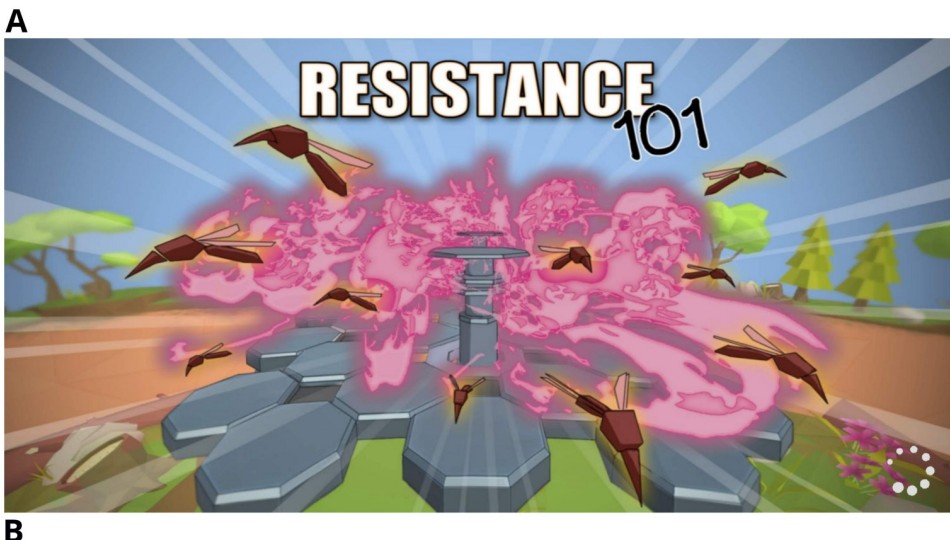

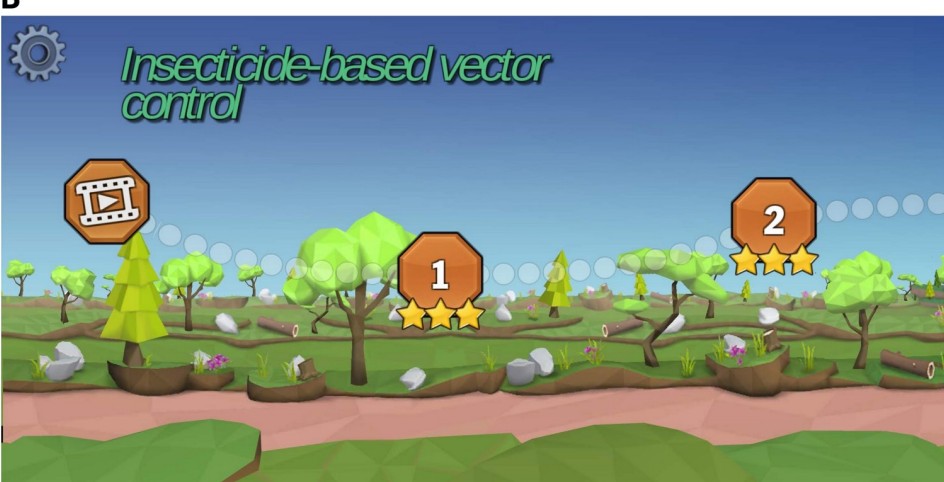

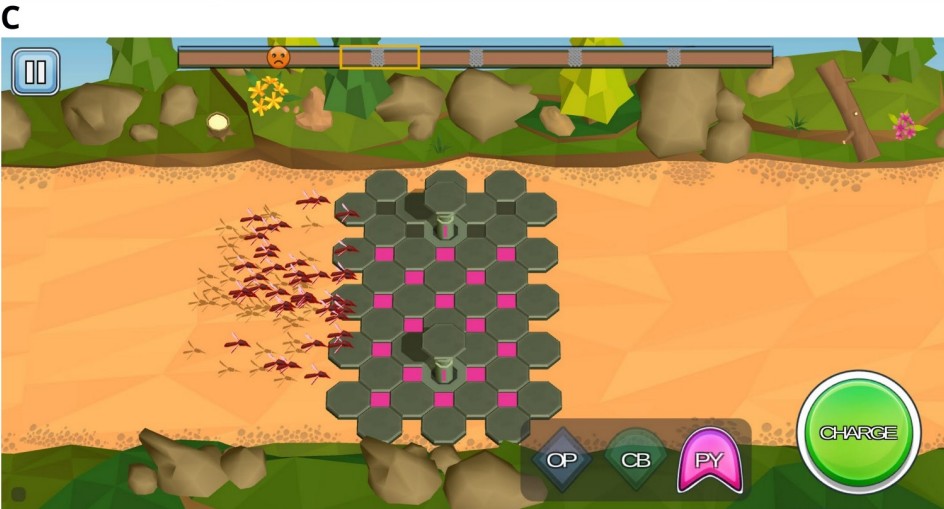

**Fig 1. Screenshots showing elements of Resistance101.** Screenshot taken of (A) the Resistance101 splash screen, (B) the main menu where players select which game level they want to play, and (C) a general representation of gameplay showing mosquitoes crossing an intervention zone where players must decide which intervention to deploy.

and managing insecticide-based interventions (16 questions connected to ResistanceSim and 6 to Resistance101) and was administered before and after the training (S1 Appendix).

Self-efficacy is "one's belief in one's ability to succeed in specific situations or accomplish a task" [13, p. 155]. Self-efficacy is typically assessed by asking participants to rate on a Likert scale how strongly they believe they can perform specific tasks. We developed a self-efficacy assessment tool containing 11 tasks focusing on key IRM areas in accordance with accepted guidelines [14] (S2 Appendix). The self-efficacy questionnaire was administered before and after the training. To evaluate if participants' attitudes towards games changed after the training, we asked participants to rate on a seven-point scale, if games could be used as a learning tool for IRM.

We developed an interview guide to assess the training itself, as well as the impact of the training on participants' work. Interviews were centred on five major open-ended questions: 1) Can you please tell me a bit more about your job? 2) How, if at all, did the topics covered within the training relate to your job? 2) Can you try to give an example of an occasion, if any, where you acted differently in your work, after the training? 3) Was there any aspect of the training which was particularly useful? If so, which one and why? 4) Was there an aspect of the training which was less useful than others? If so, which one and why? 5) Did you play the game (s) after the training had finished? Why/why not?

Ethics approval was obtained from the Liverpool School of Tropical Medicine, Research Ethics Committee (17–004), as well as from the Institute of Health IRB at Jimma University in Ethiopia (IHRPG/8/3/17) and the ERES Converge IRB in Zambia (2017-Jul-011). Written informed consent was obtained from all participants.

**Data collection.** At the start of the training, we collected socio-demographic data and information on participants' gameplay patterns (how often they play and which games) in the form of questionnaires. Participants completed the knowledge test and self-efficacy assessment tool at the beginning and end of the training. We conducted semi-structured interviews with participants one month after the training. Of the forty-eight participants in the training, thirty were available for the interviews. Interviews were conducted by one of the co-authors (CD or KD). Both interviewers had been trained in qualitative research and interviewing techniques, and CD had prior experience conducting similar research. Each interview was held in a private meeting room with just the interviewer and the participant. Interviews lasted approximately one hour, were audio-recorded and then transcribed verbatim. Notes were taken by the interviewer when feasible while still allowing them to concentrate on the conversation. Two participants were excluded from the analysis due to poor quality of the audio recording.

**Data analysis.** We compared knowledge test and self-efficacy scores before and after participation in the course using paired t-tests. If multiple comparisons were made, we used Bonferroni-adjusted significance thresholds. Where tests and rating scales were not completed in entirety in both pre and post course, participant answers were excluded from the analysis.

Guided by an inductive thematic analysis approach, interviews were coded following the guidelines from Braun and Clarke [15], which provide comprehensive steps for researchers to describe, organise and report patterns in qualitative data. This approach, based on the emergence of meaning, is very useful for capturing the richness of the data, such as nuances of participants' experiences and perspectives.

We first familiarised ourselves with the data by reading all transcripts and noting ideas and observations. We then began generating codes to help organise and interpret the data. The first four interviews were coded and discussed by two researchers (CD and KD) to establish a common understanding of the coding framework. All the interviews were then coded by a researcher (CD) using NVivo version 11 (QSR International Pty Ltd) as a supporting tool. Codes were reviewed iteratively as new ones emerged. Duplicate codes were eliminated, and

codes were deleted or regrouped as necessary. During this interpretation process, we focused on what participants said explicitly, but also looked for implicit meanings. We reviewed the data iteratively to make sure that emergent codes were applied consistently.

We then grouped the codes into categories. Maps of categories were drafted to support this process and to refine categories. A final analytic step involved searching for and defining themes and sub-themes. The data were systematically reviewed to ensure that each theme and sub-theme were identified correctly. Considerable effort was made at each stage to ensure that the emerging analysis came from the available data and reflected the experiences of the participants. The researchers strived to stay open-minded and free of preconceived ideas. Furthermore, reflective notes and memos were written and refined at every stage of the process to support the interpretation of data and emerging patterns.

# Results

## Participant characteristics

Demographic information for participants is indicated in Table 1. Participants were all educated professionals, nearly all having university education with 30 at postgraduate level. In Zambia, five participants had a technical college education instead of a university one. Participants had diverse roles in the field of vector control, from operational control personnel to academics from institutes that provide technical support to the malaria control programme. Participants had limited experience in playing digital games: a quarter did not play them at all or rarely, while a quarter played regularly. The range of games played was quite limited, including a few social games and war games, word games, quizzes, and brainteasers (S3 Appendix).

## Knowledge acquisition

The knowledge tests were scored by giving 1 point for each correct answer, allowing a maximum of 22 points per test. Forty-four participants (22 per country) completed the knowledge test both before and after the training (S4 Appendix). The results from the tests indicated a significant increase in test scores after completing the training ($p<0.001$, Fig 2).

To investigate the relationship between knowledge acquisition and the level of pre-course knowledge, the participants were ranked according to their results in the knowledge test before the training, and subsequently divided into four quarters of equal numbers of participants. There were eleven participants per quarter, Q1 represents the lowest scores and Q4 the highest scores. There was a highly significant increase in the score within the first quarter ($p< 0.001$) and a

**Table 1. Demographic information from study participants.**

| Country | Age | Females | Males | Unspecified |
|---------|-----|---------|-------|-------------|
| Ethiopia | 18–24 | | | |
| | 25–34 | 2 | 10 | |
| | 35–44 | 1 | 4 | 1 |
| | 45–54 | | 6 | 1 |
| | ≥55 | | | |
| Zambia | 18–24 | 1 | | |
| | 25–34 | 4 | 3 | |
| | 35–44 | 1 | 8 | |
| | 45–54 | 3 | 2 | |
| | ≥55 | | 1 | |
| | **Total** | **12** | **34** | **2** |

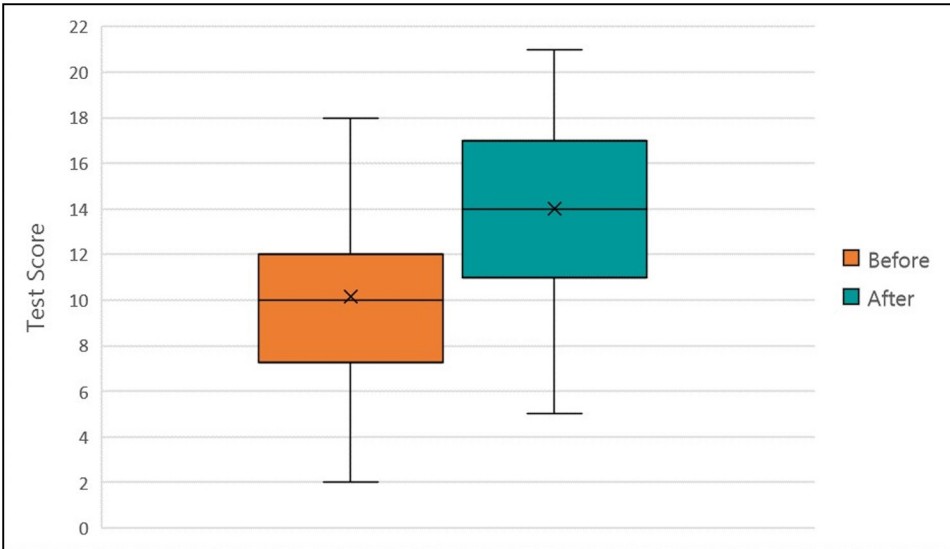

**Fig 2. Distribution of results for the participant knowledge test on IRM.** The mean score (x), 1st and 3rd quartiles (box), and range (whiskers) of pre- and post- course knowledge tests from 44 participants.

significant increase within quarters two and three (p = .005). The fourth quarter, however, showed no significant increase in score (p = 0.020 with a Bonferroni-adjusted alpha of 0.0125, Fig 3).

## Self-efficacy

A total of 40 participants completed all questions in both the pre-and post-training self-efficacy assessment tool (S5 Appendix). Self-efficacy was scored as the sum of the Likert points for all questions. The mean score of all participant self-efficacy questionnaires showed a highly significant increase after the training (p<0.001, Fig 4).

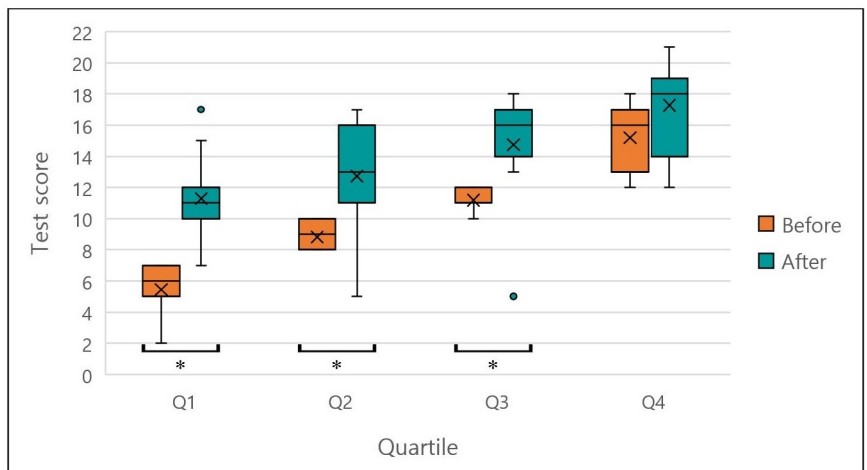

**Fig 3. Distribution of results for the participant knowledge test on IRM by pre-course test performance.** The mean score (x), 1st and 3rd quartiles (box), and range (whiskers) of pre- and post- course knowledge tests from 44 participants. The data was divided into 4 groups (Q1-Q4) based on scores attained in the pre-course test. Asterisks indicate significant differences at a Bonferroni adjusted alpha of 0.0125.

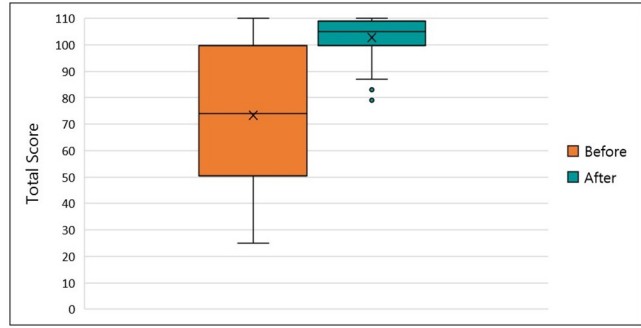

**Fig 4. Distribution of results from participant surveys on self-efficacy in vector control and insecticide resistance management.** The mean score (x), 1st and 3rd quartiles (box), and range (whiskers) of self-efficacy scores before and after participation in the course (n = 40).

To investigate the relationship between change in self-efficacy and the level of pre-course self-efficacy, participants were divided into four quarters, according to their results before the training. There was a highly significant increase in the score within the first three quarters, $p < 0.001$, with changes in self-efficacy most pronounced in individuals who scored in the first quarter during the self-efficacy pre-test (Bonferroni-adjusted alpha of 0.0125, Fig 5). The fourth quarter showed no significant increase in score, $p = 0.706$. The magnitude increase in self-efficacy scores was consistent across all 11 questions of the questionnaire.

## Participant perceptions

Before and after the training, participants were asked if they thought digital games could be used 1) as learning tools, and 2) to convey best practices in vector control. They expressed their level of agreement on a 7-point Likert scale. After the training, participants more strongly agreed that games could be used as learning tools (increase in the mean ± SEM from 5.95 ± 0.185 to 6.95 ± 0.034) and to convey best practices (5.35 ± 0.191 to 6.85 ± 0.076) ($p < 0.001$ for both comparisons, S6 Appendix).

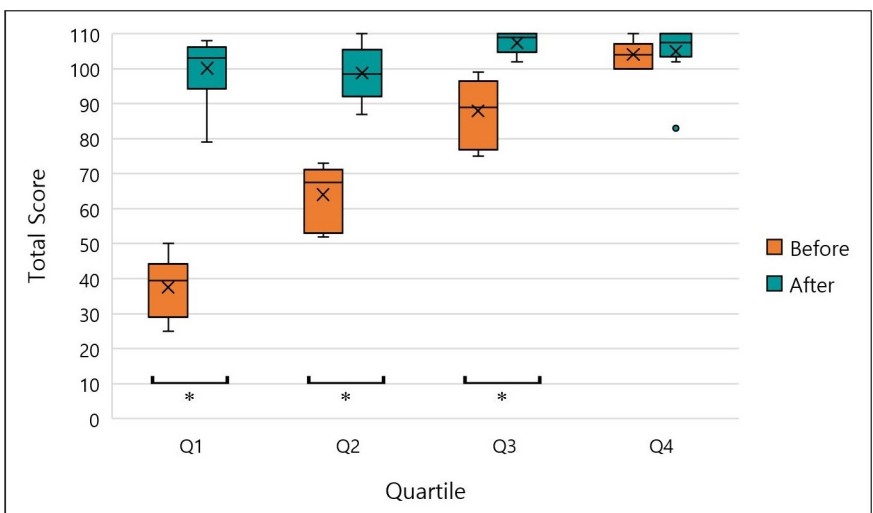

**Fig 5. Distribution of results from participant surveys on self-efficacy in vector control and insecticide resistance management by pre-course self-efficacy.** The mean score (x), 1st and 3rd quartiles (box), and range (whiskers) of pre- and post- course self-efficacy scores from 40 participants. The data was divided into 4 groups (Q1-Q4) based on scores in the pre-course evaluation. Asterisks indicate significant differences at a Bonferroni adjusted alpha of 0.0125.

## Qualitative thematic analysis

We present the results of the inductive thematic analysis based on the participants' interviews. An overview of the themes and sub-themes are first given in Fig 6, and subsequently discussed.

**Theme: Game-based training.** Participant interviews provided insight on participants' reactions to and perceptions of three main areas: the use of games as training tools, the training,

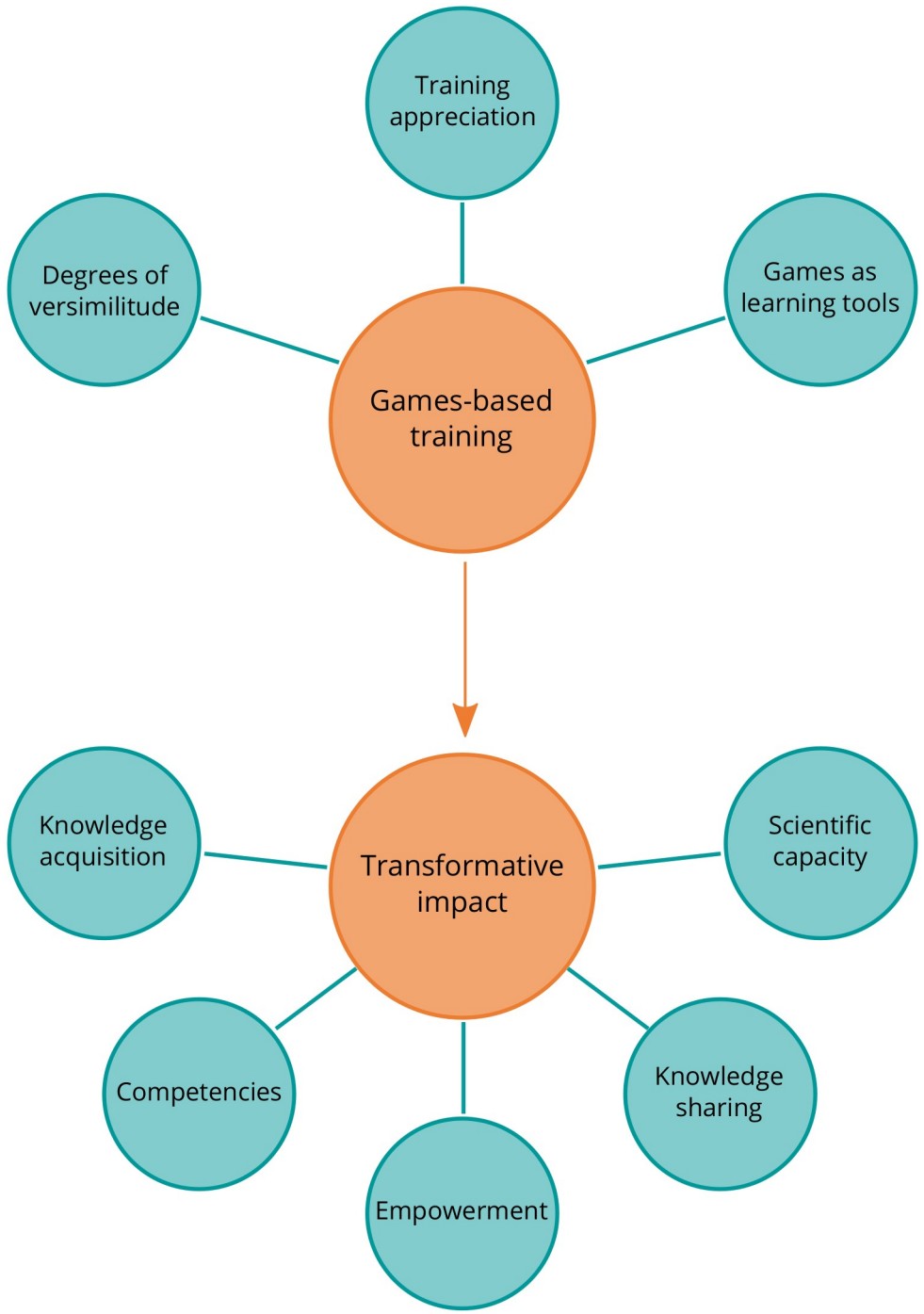

**Fig 6. Overview of the thematic analysis.** Themes are shown in orange circles, and sub-themes are shown in turquoise circles.

including perceived strengths and weaknesses, and verisimilitude of ResistanceSim compared to IRM in the field. Participants' reactions to the training and the games were overwhelmingly favourable, and participants used a range of positive affective terms in relation to both the games and the training.

*Sub-theme*: *Games as learning tools*. Participant reflections on the games and their experience included what they learned and an awareness of the specific game characteristics that supported their learning. Participants' attitudes toward the use of the games as a learning tool were enthusiastic, expressing that they enjoyed playing the games. Besides being enjoyable, participants described the games positively, primarily in two emotional dimensions: pleasantness and excitement, using descriptive words like "friendly", "nice", "good", or "interesting", "stimulating", "exciting".

Their game experience with ResistanceSim and Resistance101 was distinctly different. One participant noted that Resistance101 was more like a game while ResistanceSim was more like a professional tool. Resistance101 was liked by all, and strongly seen as a "play and learn" tool, a simple game with strong clear mechanics, "choosing the insecticide and shooting all the mosquitoes". Participants mentioned the animations of Resistance101 were enjoyable, and a good way to learn through presentation of concepts visually and vividly: "the video is easy to grasp compared to just lectures." One participant stressed how he could visualise the biological aspect of resistance:

> "To just learn about how the neurons work and you are looking at the pictures and you are seeing what is happening, the positive charges, the negative charges and all that, it really helps one to understand."

Participants saw ResistanceSim as more complex and demanding, but interesting and motivating. The game required more concentration and problem-solving skills.

> "You have to play with different parameters. There are insecticide of different classes that are acting on different ways, so that part impressed me and it's like playing a game, like when you are playing chess you have to think in many ways."

The engaging nature of the games and their efficacy as learning tools can be explained by several game characteristics like scaffolding, feedback and replay value. Participants reported that the sequential structure of Resistance101 and the ResistanceSim RoadMap facilitated the acquisition of knowledge. Participants praised the progression of Resistance101, including the sequential introduction of scientific concepts in each animation, as well as levelling up during gameplay (going from the easiest to the most complex). Participants mentioned feedback as a positive feature of the game. Feedback is given in the game in tutorials, during gameplay, and after failures. Participants felt that visual and vivid qualities of the games facilitated retention and allowed flexibility and freedom in learning, which participants mentioned: "learning at your own pace", or allowing you to "know every detail" and "I repeated the game several times to capture the science as well as the practical things."

The participants recognised the key messages from the game-based training. The game mechanic in Resistance101, where the player has to shoot mosquitoes with the right insecticide at the right time, was notable in effecting participant understanding.

> "Because it always also shows you that if you do an intervention at the right time with the right dosage and the right chemical, you're going to get a proper result and you won't have that resistance."

Although participants acknowledged the value of continuing to play after the training, half of them mentioned that finding the time to do so was a major issue.

"That game is not just actually playing and making fun. Yeah, in the game, the information behind the game is very useful and that one is an actual target for this course. So, having the time is one of the key factors."

Participants described having many demanding tasks and activities, and some participants were often out of their offices conducting field work. Network and infrastructure limitations were also mentioned, including access to and availability of PCs. Mobile phones were clearly the preferred platform.

*Sub-theme*: *Training appreciation*. Participants reflected more generally on the training as a whole, rather than just the games themselves, by comparing the game-based course with other courses, highlighting its strengths and weaknesses. They assessed the training in slightly different ways than the games themselves, referring to coverage and content in terms such as "comprehensive, interesting, useful and helpful". Participants differentiated the course from previous courses by referring to it as "unique" and "awesome".

"For me? It was the first time I think I attended such a training which was interesting from the beginning to the end."

Participants compared the game-based training to traditional teaching, based on "presentation and PowerPoint". As someone stated, "he preferred to interact than to read". Participants appreciated the training structure and flow, mini-lectures, gameplay, discussions and the group exercise. They placed high value on the practical or hands-on aspects of the training embodied through the games. Participants contrasted the training's theoretical and practical aspects. Participants were first introduced with information on IRM concepts and processes, which was followed by their immediate revision and application through gameplay. Participants found the group work, and the coaching during the gameplay sessions, very helpful, as they were monitored to observe progress and help was provided as needed.

Participants specified that the whole training was relevant to them and increased their knowledge, and they also stressed that the training was directly related to their job: "is all about insecticide management, resistance management, it all involves our job".

Despite the success of the training, some aspects need to be strengthened. According to participants, the training was too short, too fast, and there was not enough time for participants to play the games. Participants felt they were rushing too much between the training components, but most of all, participants wanted more time to play, especially for ResistanceSim.

"Playing ResistanceSim unfortunately for me, I really needed much time on it, because it is a game alright, but it is more a technical part than just a game".

This is quite understandable, participants first needed enough time to become familiar with the game environment and tasks, and then to assimilate the different concepts and procedures. This is especially true for a multi-layered simulation game like ResistanceSim. Consequently, as a participant highlighted, it might be difficult to re-engage with the game if they had not mastered the game well enough during their initial exposure.

Some participants requested additional support with the game-based training, especially reading material, a game manual / documentation, and feedback on their progress at the end of the course. Participants also made several suggestions to make the training more effective,

consolidating it by adding follow-up components such as reminders to play, incentives to boost their interest such as an online quiz, and new versions of the games. Participants also agreed that a short refresher training could be useful:

> "Maybe, after six months so that we review and see how the course has impacted on our work. Or maybe again, as a reminder. So that we play the games and then get to know the games better."

Finally, some participants requested adding a field component to complement the course, going over some of the steps from the game-based training, to experience in real-life those aspects covered in the game and for problem-solving in real situations, as needed.

*Sub-theme*: *Degrees of verisimilitude in the games*. Resistance101 is an introductory educational game, while ResistanceSim is a simulation game. Participants noted that current operational settings were not as ideal as those depicted in the games, especially in ResistanceSim, as not all options are available to them.

> "Yeah. Because the differences are that in IRM we are trying to equate the theory of the game, the application and then also the changing the chemicals, rotating them and rotating more of the chemicals than just coming from the centre. We don't have those options at the same time."

More than half of the participants mentioned resources (budget, human, products and material) as the most important barrier to the success of IRM. Limited budgets have a strong impact on the choice and quantity of tools for vector control. The logistics of IRM can also be a challenge including transportation, equipment maintenance, or delivery of chemicals on time. To compound this, access to scientific equipment, "inadequate tools to use", might be lacking.

> "For example, we know that morphological identification, the PCR identification is much better as a morphological identification, but changing this will require additional resources, so we are using, we are just testing samples, subsamples of the mosquitoes that we collected from the field with the PCR."

Some of these operational challenges were embedded in the game mechanics: a set budget restricts the strategy that players can implement e.g. a limited amount of insecticide/bed nets can be procured. In addition, the efficiency of some operations in the games, such as spraying, was decreased by lack of adequate training. Potential future versions of ResistanceSim could include other operational barriers to reflect conditions in a particular region or country.

**Theme: Transformative impact.** The training increased participants' knowledge of insecticide resistance, and procedural knowledge of IRM. The game-based training seems flexible enough to accommodate a range of needs, filling knowledge gaps, providing detailed information, or refreshing their knowledge.

*Sub-theme*: *Knowledge acquisition*. Many participants needed to gain new knowledge in insecticide resistance or IRM. One participant stipulated he had previously found it difficult to understand some of the issues when he had tried to self-learn. Some participants had moved from other specialisations and were missing expert knowledge in some areas, in particular in the area of entomology. Due to the nature of their jobs or / and because some procedures are centralised, participants were also lacking vital information.

"I would think maybe basic susceptibility tests like I said, I never really observe how it's done but the course was able to bring that one out." "So, it can guide me to have the appropriate tool for surveillance, to use the appropriate technique for species identification."

Some participants at the district level stated that they appreciated getting insights on "what happened at the higher level", as they are not involved in the choice of insecticide. The training assisted meaning-making: it helped participants to integrate knowledge or information and gain new insights that enhanced their understanding.

"Like I got a grip of it, I could understand what they really mean when an insect is, like the mosquitoes are resistant to a specific type of insecticide and what not" but he added that "this course adds a lot of things to me, because you see, I learnt these two areas in a way that can be conceptualised".

Some more experienced participants felt that the training refreshed knowledge and skills that they had not used for a while. One participant summed this up as "Okay the course that I took with you has mostly, kind of, enhanced the knowledge that I had".

The game-based training, and especially Resistance101, gave participants an opportunity to get an overview of insecticide resistance, and fill gaps in their understanding. Many participants highlighted the specific topics they learned such as insect biology, mechanisms of insecticide resistance, insecticide modes of action, as well as strategies to manage resistance.

"It is changed positively in that I now fully understand why it is important not to use one type of chemical for IRS and how cross-resistance works."

Participants noted that the course information was directly applicable to their jobs. As a participant emphasised, they work with insecticide, and they need to know how to strategically use it to prevent resistance from developing, "it was very useful actually, as we know, IR is very problematic and a barrier to malaria elimination". As a participant noted, knowledge of insecticide resistance will enhance their job performance, "know how this works, how I will just apply this and it will work".

ResistanceSim enhanced participants' procedural knowledge. Some participants indicated that the game provided a good overview of how to manage a vector control programme and guided them through the steps of the IRM process. This overview consolidated participants' understanding.

"I found the RoadMap interesting because it was a map, it was like a process on vector management, so you are going from this step to that step. So even as you are doing your work, you are supposed to start from here, collection of data, so I found that very interesting."

Quite a few participants lacked knowledge of advanced techniques. As a participant noted, he now knew the purpose of a synergist assay and which steps to follow to apply it effectively.

"Even the advanced procedures also mentioned in the course, you know the PCR, the synergist also, which is not implemented even in this country and . . . but we didn't implement the synergist form of the insecticide here. But I have got the detailed information from the course."

*Sub-theme*: *Competencies*. The training enhanced participants' understanding of systematic decision-making at different stages of the process. Participants recognised the importance of

making evidence-based decisions, and thus collecting appropriate and quality data at key points. One the most important outcomes of the game-based training course relates to a greater realisation that the quality of the intervention was key to the effectiveness of IRM. Consequently, it led to some changes in work practices.

Participants acknowledged the importance of data collection to inform IRM. The game-based training highlighted the importance of collecting baseline data and monitoring interventions. After the training, participants reiterated a key point from the course that collecting baseline data will help participants to make comparisons after an intervention and make data-driven decisions to ensure effective IRM. A participant elaborated:

"The same species can act differently in different geographical areas. So, based on your baseline data you need to choose appropriate insecticide or intervention in general for which species, for which study sites and then the likes. This is what I learnt from the programme. Yeah."

The importance of monitoring was highlighted as follows:

"So, it's better, according to this expert training, according to this part it is better to monitor that part of the intervention regularly. So, if there are any matters of resistance it is better to look for another option either with integrating another intervention or using techniques like mosaics, rotation, mixture of the insecticide."

For some participants, the training changed their outlook on IRM, they now recognise the importance of making evidence-based decisions "I need to look at my data first before I make a decision." The training led to differences in the decision-making process and resulted in changes in insecticide spraying strategies.

"And those were the ones that had the most occupying malaria incidences. So, that was informed by the training where we said okay, let's look at the baseline data. We just don't say look what's the structures. The structures in this and how many you target like that, but you say no, let's record the incidence and I think that's what will inform us where we will spray. Even within the hotspots we had less chemicals. We had to request for some more, about 862 bottles extra and they extended our spray number of days from 17 to 25 days and then I think we covered. So, we looked at the baseline information that was existing to make decisions. I think that also came from the course."

The game-based training helped learners to visualise what will happen in the field from the operations and decisions that are made and thus can have some practical implications on their work. ResistanceSim and the game-based training more generally were seen as tools that could support decision-making.

"Oh yes. I do think the course can make a difference, yes, encourage our decisions and can even improve the way programmes are being done. Yeah. So that we make decisions based on scientific data that on the risk points of the vector to whatever type of insecticide the characteristics of the vector it can only help in changing a lot of things."

Not only did the training improve their decision-making, but it appeared to motivate participants to raise the quality of the insecticide-based intervention.

"This programme can tell them to follow the right protocol, to follow and use the right tool for monitoring, to follow the standard operating procedure, to have good quality data as well as to apply the appropriate intervention."

Participants reflected that resistance could be caused through their negligence, or misinterpreting instructions or information. For example, IRM is not always fully implemented or IRS conducted adequately "missing hotspots with high cases of malaria".

"Before we were just saying okay, if one could use a mosquito net then they could as well substitute that net with the spray, so it's either this or that. Whereas this time we understood that we needed to combine. If we just say okay, I have a mosquito net, I don't need the spraying, then we are kind of promoting resistance because we are just using one insecticide, so that's how it's actually connected very well."

Thus, the game-based training also helped participants to recognise some flaws in their actions or reasoning thus helping to correct them, "to have the right thing done, so that can help us to see where we are going wrong". Quite pragmatically, as a participant added, if they do the wrong thing, they will not get any results, so they may as well do the right thing.

*Subtheme*: *Sharing knowledge*. Sharing knowledge was one of the primary actions that participants reported taking. Indeed, participants recognised the importance of the knowledge they gained and its impact on IRM. One participant mentioned, "When people have knowledge, they act different".

Participants better appreciated the importance of involving and exchanging information with major IRM stakeholders, including policymakers dealing with malaria elimination, "how we can seek assistance or advice from other ministries". Participants felt that a greater cooperation was needed with everyone concerned: trust would be secured by informing stakeholders, establishing consensus about decisions, and fulfilling responsibilities as planned.

"All the district heads of departments were invited to a meeting and then also other key stakeholders and we ensured that they came. But previously, maybe we would have just got whoever came and we went ahead. By this time, we wanted to make sure that we explained this to all at once and then when we came out of the meeting, whatever we agree is what is going to be done. Yes. And we stuck to the dates. When we say we are going to spray this for maybe three days, we would be there three days, this facility four days, it would be four days as agreed. So, I'm sure even the stakeholders were pleased."

Several participants told us that they had or would share knowledge with their closed social networks, primarily colleagues in similar positions, but also subordinates and supervisors. Besides knowledge, participants also mentioned wanting to share the games with their closed and specialist networks like the Ethiopian Malaria Research Network, or at malaria workshops. Note we specified at the time that participants could only share the games once they were officially released.

"I think it would really help so I was thinking of the officers in the district. I think it would be helpful to introduce the game to them and make sure they know how to play them. People would be able to build on their own knowledge and capacity in that way"

Training seems quite a regular activity that at least half the participants are carrying out. Because of a high turnover of personnel, the most common training related to insecticide

resistance and spray operations. After the course, training field staff adequately and more thoroughly was seen as quite important to improve the quality of the intervention.

> "They will tell me just look at the odd spray operators and just go into the field for a day or two then . . . but this time I acted differently because I refused, you are doing shoddy work. They will just tell you do two days and then go and . . . I refused this time, so I did adequate training and the job is . . . I think it's going one better than before."

Participants identified that existing training of spray operators did not include much about insecticide resistance, so participants started sharing knowledge of IR with them and noted that it should become integral to part of their training. Participants felt that they too should understand the importance of doing the right thing at the right time, and the consequences of not doing it. "So it gave them some insight, some ideas, for them to do the correct thing if we are to prevent malaria." By doing so, the operators can also feel more valued, as they see the importance of their role. It would also allow spray operators to communicate better with communities, gaining their trust and cooperation.

Community issues in IRM are well-known including bed net use, and acceptance of IRS spraying. There was a strong feeling from participants that a greater social mobilisation was needed with communities, otherwise "there would be no solidarity." Thus, informing and sharing knowledge with the community was seen as quite important, as it could affect engagement and cooperation, "and the fight now becomes their fight". There was a realisation that they in turn would benefit from knowing the scientific evidence about insecticide resistance, especially community health workers, acting as intermediaries between communities and malaria programme managers.

> "If you know nothing about vector management, you won't even know the importance it has in malaria elimination. So at least it brings about awareness and you also feel like you are part of the team."

The training also provided participants with the information to effectively communicate with communities to increase acceptance of IRS.

> "But you just gave us mosquito nets. Why are you spraying again?" So, we acted differently by saying, what we are doing is we are trying to manage resistance. Previously, I think I would have had no proper reason to give, but this time I was able. It was just after the training; the spraying came, and we were able to explain like that."

*Sub-theme*: *Facilitating empowerment*. The game-based training seems to have empowered vector control professionals. Besides gaining new knowledge and enhancing competencies, the training also increased participants' motivation, confidence and dedication to overcome IRM challenges. The training increased participants' motivation for their work, as they learn how to overcome the many challenges that they are facing. One participant mentioned, "I got more interested in doing what I was doing," and another that he would continue learning:

> "Therefore, the point that you have raised encouraged us to know in detail insecticide resistance management is very important for our country".

The training also reinforced their motivation to engage in problem-solving and learning from their experiences:

"What we learn is just to prime us, to prepare us to what we are going to find in the field and it's up to you to develop that capacity to overcome what you want. Because not everything we are going to be taught anyway during the training. But it is your drive and your inner motivation which is going to make you like everything."

The training raised participants' confidence, as they were given the tools to confidently share their expertise and participate in meetings more authoritatively, thus raising self-esteem.

"So, my experience with the training was able to comment in that particular meeting. I think I was speaking more with some authority. I understood what I was talking about and what the other people were talking about."

Based on his new understanding about entomology, a participant felt better integrated in his professional field, as he was now able to participate in discussions. Similarly, others mentioned being able to share knowledge, inform and advise others as "when you're speaking from an informed point of view". Thus, participants felt more empowered in their work; it is not just the policy makers that can tackle that issue, but everyone can do something at their level. By making insecticide resistance and its management clear, the training stimulated action to sustain malaria control.

"I appreciate this innovative approach in making resistance very clear to people and encourage people to take action and contribute for resistance management strategy. One thing that might be a priority in the future."

Empowerment in this context also means the participation of malaria professionals in the IRM decision-making process, but some participants highlighted that this was not the case. National strategies were not always seen as effective, as the situation in the field varies greatly from one end of the country to the other.

"If we can establish entomological and insecticide resistance studies and monitoring at district level, it would help us, because we would be sharing the scientific evidence, finding out what is the best insecticide to use, and what to avoid, as opposed to what is happening now"

Decisions regarding the choice of insecticide could be influenced by the private sector, which might not be in line with the national strategic plan, or by economic criteria "cost effectiveness, might be more important than scientific evidence". The lack of involvement of malaria professionals in the decision-making process could render IRM less effective.

*Sub-theme*: *Supporting scientific capacity*. As a participant noted, the success of the game-based training, and its impact on IRM and malaria control in general, can only occur if knowledge is propagated. Not only should knowledge be shared in an operational context, but the training should be widely distributed.

This participant suggested that the training be disseminated at national and regional levels, strengthening the capacity of all malaria professionals including programme district managers, environmental health officers, and related government staff, especially policy makers.

"I have a team of environmental health officers who help supervise the spray operators so they need to understand."

Several participants mentioned that the training should be given for newcomers, everyone coming to work at the malaria programme. It should also be given to students in colleges and universities, especially new graduates involved in research. It was also suggested that all partners and stakeholders involved in the malaria programmes would benefit from the training, such as those in the private sector and medical staff at health centres treating malaria cases. Indeed, medical staff are in a unique position to raise community awareness. One participant also suggested involving traditional healers.

"Because those are the people who will help us in common behavioural change management communication, for them to accept the programme".

Participants also advocated that the game-based training, or some of its components, suitably modified, should be used as part of community sensitisation and mobilisation.

"But if they were able to work through that and understand the whole process, that just makes life easier for all of us and that even motivates them".

## Discussion

Our findings suggest that educational games can be powerful instruments to convey scientific knowledge and change operational practices in the context of IRM and malaria control and elimination in Ethiopia and Zambia. The digital game-based training in insecticide resistance and IRM facilitated knowledge transfer, narrowed knowledge gaps and equipped participants to alter their professional practices, thus encouraging sustainable implementation of IRM.

The success of the training can be explained in part by the choice of the interactive game media and the game design strategies, and in part by the game content. Affective-motivational states, as they occurred in the training, play important roles in learning contexts [16]. Participants embraced the games and their emotional reactions were overwhelmingly positive. Emotion influences various cognitive processes that are involved in the acquisition and transfer of knowledge and competencies [17]. Indeed, interactive and engaging interventions such as the game-based training are known to be more effective in knowledge transfer than passive ones [18]. The impact of the games was consolidated by the overall training strategy, blending learning through mini-lectures and gameplay, weaving between personal coaching, discussions and group work.

The knowledge tests indicated that participants developed, consolidated and enlarged their knowledge base, but improvement was variable. Understandably, participants with the highest level of knowledge prior to the course did not appear to benefit much from participating in the training on this dimension. Increases in knowledge test scores immediately post-training indicate that this course could be most beneficial in training individuals with less IRM knowledge. It would be important to evaluate if gains in knowledge acquisition by participants with little previous knowledge translate into application, as it is not known if their increase in knowledge would be enough to successfully carry out IRM tasks. The thematic analysis emphasised some of the scientific and procedural knowledge that participants gained and valued, but further assessments would be needed to show if some of the course content needs to be strengthened. The game-based training took place over a limited period—two-and-a-half days. Allowing more time for gameplay and consolidation, as suggested by the feedback, could allow those participants in the lowest quarter to benefit more from the training and increase their performance. Alternatively, it may be beneficial to deliver this course to participants grouped in cohorts based on knowledge test scores. It would also be beneficial to establish some basic

knowledge acquisition standards: what level should be achieved in which professional context, and what knowledge participants should have before the training to maximise proficiency.

The successful acquisition of knowledge does not conclude the process of knowledge transfer [19]. Knowledge transfer and its impact can be assessed through three outcomes: (a) knowledge acquisition (b) changes in attitudes, and (c) changes in practice [20].

A change in attitudes has been defined in this context as valuing the acquired knowledge with the motivation and sense of self-efficacy to use it [21]. As we have seen, there was a significant increase of participants' acceptance of educational games to convey best practices in vector control. Participants appreciated the value of the game-based training to fill knowledge gaps and support their work. The importance of motivation as a benefit from the training should be emphasised, as it improves professional effectiveness and facilitates changes in work practices [22].

The self-efficacy test showed a marked increase for all participants in the first three quartiles. The training did not have an impact on this dimension for those already highly confident. As we have seen in the thematic analysis, participants felt empowered to share their expertise and knowledge. Indeed, studies have shown that self-efficacy enhances information sharing [23]. Although numerous studies have found a positive relationship between self-efficacy and performance, this is not always the case, especially over time [24]. After the training, the self-efficacy score became quite high for almost all participants regardless of their knowledge scores, so there is some danger that some participants became overconfident. Thus, self-efficacy and its factors should be further investigated in relationship with the training design, and its longer-term impact on work practices.

Empowerment in the workplace has been defined as enhancing motivation and feelings of self-efficacy, as well as skill enhancements and authority to participate in decision-making [25]. As shown, the game-based training seemed to have increased participants' feelings of empowerment. However, delegation of decision-making within regions, and a greater participation in national strategies, could strengthen the participants' involvement in IRM, and thus the efficacy of malaria vector control interventions.

Changes in work practices can be appraised through actions undertaken so they become more efficient and effective. The thematic analysis highlighted some changes in behaviour, for example in using evidence from baseline data to inform operational management of IRS. The analysis also showed a change in attitude regarding ensuring quality interventions, and consequently some of the steps that should be taken to enhance quality. Other examples related to knowledge sharing.

Knowledge sharing is also an important component of knowledge transfer. In this study, knowledge sharing was a salient component of participants' behaviour (or intended behaviour) after the course. Knowledge sharing occurred during communication with colleagues, collaboration with stakeholders, and spray operator training. Through engaging and building good relationships with others, knowledge sharing enhances confidence and trust.

An interesting development from the game-based training is its potential to enhance social capital in relation to malaria vector control. The training facilitated interpersonal connections, and therefore has the potential to enhance collaboration. After the training, participants discussed the necessity of establishing a common ground with shared values, reciprocated understandings, and acknowledgment of the importance of each stakeholder's contribution. Thus, the game-based training could stimulate cooperative relationships between malaria professionals to collectively resolve problems [26]. The impact of the game-based training could be maximised by dissemination at national and regional levels, and then by consolidating networking and knowledge sharing, thus making some basic knowledge about IRM accessible to all.

To become an important tool in the fight to control and eliminate malaria, the game–based training and supportive didactic material should be embedded in a collaborative training platform, offering scalable up-skilling opportunities for professionals in vector control. Mechanisms to support knowledge sharing, social mobilisation and networking could also be provided to maximise its effectiveness. Finally, game updates to parallel recent developments in insecticides and new tools in vector control will have to be considered.

## Limitations

The first language of most participants was not English, and their level of verbal English was quite heterogeneous, as is evident from some participants' quotes. This could have impacted the interviews, and thus the thematic analysis, and possibly other aspects of the training. Getting a translator for the interviews was not feasible, for strategic, logistical and cultural reasons. However, if further studies and/or implementation of training programmes are planned, it will be important to consider the use of translators or delivery of the training in the language that participants are most fluent in. There were limitations regarding knowledge transfer, and the changes in work practices that participants could relate to due to the interview timing. Having the interview one month after the training enabled participants to recall the training and associate changes in work practices with it. However, it was too limited to fully assess the impact of the game-based training on participants' work. As participants noted, most activities are planned and there is a considerable amount of seasonal variation in the indoor residual spraying and the IRM process. Thus, complementary studies and methodologies are needed to assess changes in more depth, as well as longitudinal studies to evaluate its impact over time. A higher number of participants across several training sessions would also allow further analysis and extraction of information from quantitative data. For example, due to the small sample size, it was not possible for us to investigate differences in performance at different Bloom's cognitive levels. Furthermore, evaluations in different African countries that more comprehensively represent the diversity of educational/technological capacity, malaria transmission, and insecticide resistance status would also extend the reliability of the course as a regional solution.

## Conclusion

In this paper, we have described the evaluation of an innovative game-based training course for insecticide resistance management, centred on two games ResistanceSim and Resistance101, to build capacity for IRM in malaria control and elimination. From the results of the quantitative and qualitative analysis, it is likely that the games played a critical role in the success of the training. The evaluation results indicated that the training facilitated knowledge transfer, enhanced knowledge acquisition, and stimulated changes in work practices in Ethiopia and Zambia. The course therefore has the potential to be a transformative tool in vector control for establishing communication and encouraging good practices. Continued implementation and evaluation of the training in other contexts would help establish the robustness of the game-based training. Furthermore, developing tools to capture changes in work practices over time should be considered. Consolidation of the game-based training, and plan for wider dissemination, will see it as a tool for increasing capacity, enhancing the scientific capital of sub-Saharan African countries, and nurturing new generations of malaria professionals.

## Supporting information

**S1 Appendix. Knowledge test.**
(PDF)

**S2 Appendix. Self-efficacy questionnaire.**
(PDF)

**S3 Appendix. Participant characteristics.**
(XLSX)

**S4 Appendix. Knowledge test scores.**
(XLSX)

**S5 Appendix. Self-efficacy scores.**
(XLSX)

**S6 Appendix. Perceptions data.**
(XLSX)

## Acknowledgments

Many thanks to all participants in Ethiopia and Zambia who participated in the game-based training course and its evaluation.

## Author Contributions

**Conceptualization:** Charlotte Hemingway, Marlize Coleman, Michael Coleman, Edward Thomsen.

**Data curation:** Claire Dormann, Kirsten Duda, Edward Thomsen.

**Formal analysis:** Claire Dormann, Kirsten Duda, Edward Thomsen.

**Funding acquisition:** Marlize Coleman, Michael Coleman.

**Investigation:** Claire Dormann, Kirsten Duda, Edward Thomsen.

**Methodology:** Edward Thomsen.

**Project administration:** Kirsten Duda, Busiku Hamainza, Delenesaw Yewhalaw, Edward Thomsen.

**Resources:** Busiku Hamainza, Delenesaw Yewhalaw.

**Supervision:** Michael Coleman, Edward Thomsen.

**Visualization:** Kirsten Duda, Edward Thomsen.

**Writing – original draft:** Claire Dormann.

**Writing – review & editing:** Claire Dormann, Kirsten Duda, Busiku Hamainza, Delenesaw Yewhalaw, Charlotte Hemingway, Marlize Coleman, Michael Coleman, Edward Thomsen.

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
