## [Decision Letter · Decision Letter 0]

14 Aug 2020

PONE-D-20-12756

Evaluation of a game-based training course to build capacity for insecticide resistance management in vector control programmes

PLOS ONE

Dear Dr. Thomsen,

Thank you for submitting your manuscript to PLOS ONE. After careful consideration, we feel that it has considerable merit but would nevertheless appreciate your consideration of the points raised by reviewer 2. Therefore, we invite you to submit a revised version of the manuscript that rebuts or addresses the points raised during the review process. 

We look forward to receiving your revised manuscript.

Kind regards,

Basil Brooke, PhD

Academic Editor

PLOS ONE

Journal Requirements:

2.We note that you have indicated that data from this study are available upon request. PLOS only allows data to be available upon request if there are legal or ethical restrictions on sharing data publicly. For more information on unacceptable data access restrictions, please see http://journals.plos.org/plosone/s/data-availability#loc-unacceptable-data-access-restrictions.

3.Thank you for stating the following in the Competing Interests section:

[I have read the journal's policy and the authors of this manuscript have the following competing interests: MaC made substantial contributions to study conceptualisation. At this time, MaC was a researcher working for the Liverpool School of Tropical Medicine. Before study implementation, MaC became employed by IVCC, who supported the development of ResistanceSim. While at IVCC, MaC did not contribute to study design or implementation, and her involvement was limited to reviewing, editing, and approving the manuscript.].

Reviewers' comments:

Reviewer's Responses to Questions

**Comments to the Author**

1. Is the manuscript technically sound, and do the data support the conclusions?

Reviewer #1: Yes

Reviewer #2: Partly

2. Has the statistical analysis been performed appropriately and rigorously? 

Reviewer #1: I Don't Know

Reviewer #2: Yes

3. Have the authors made all data underlying the findings in their manuscript fully available?

Reviewer #1: Yes

Reviewer #2: Yes

4. Is the manuscript presented in an intelligible fashion and written in standard English?

Reviewer #1: Yes

Reviewer #2: Yes

5. Review Comments to the Author

Reviewer #1: I have no specific comments to make. The response to the above item 1 should be "I don't know". This is not my field of expertise. Item 3 - yes, the data appear to be available, and Item 4, yes, the manuscript is well written and easy to read.

Reviewer #2: Detailed comments for each section of the manuscript are appended with this review

6. PLOS authors have the option to publish the peer review history of their article (what does this mean?). If published, this will include your full peer review and any attached files.

Reviewer #1: No

Reviewer #2: **Yes: **Givemore Munhenga

---

## [Author Response · Author response to Decision Letter 0]

7 Sep 2020

Thank you for providing valuable comments on this manuscript. We have addressed all of your concerns and have provided a detailed description of changes made in the "Response to reviewers" document attached to this submission.

---

## [Editor Report · Decision Letter 1]

29 Sep 2020

Evaluation of a game-based training course to build capacity for insecticide resistance management in vector control programmes

PONE-D-20-12756R1

Dear Dr. Thomsen,

We’re pleased to inform you that your manuscript has been judged scientifically suitable for publication and will be formally accepted for publication once it meets all outstanding technical requirements.

Kind regards,

Basil Brooke, PhD

Academic Editor

PLOS ONE
---

## [Editor Report · Acceptance letter]

5 Oct 2020

PONE-D-20-12756R1 

Evaluation of a game-based training course to build capacity for insecticide resistance management in vector control programmes 

Dear Dr. Thomsen:

I'm pleased to inform you that your manuscript has been deemed suitable for publication in PLOS ONE. Congratulations! Your manuscript is now with our production department. 

Kind regards, 

on behalf of

Dr Basil Brooke 

Academic Editor

PLOS ONE